# The Effect of Phosphate on the Activity and Sensitivity of Nutritropism toward Ammonium in Rice Roots

**DOI:** 10.3390/plants11060733

**Published:** 2022-03-09

**Authors:** Kiyoshi Yamazaki, Toru Fujiwara

**Affiliations:** Graduate School of Agricultural and Life Science, The University of Tokyo, Yayoi, Bunkyo-ku, Tokyo 113-8657, Japan; atorufu@mail.ecc.u-tokyo.ac.jp

**Keywords:** nutritropism, environment, nutrient interaction, desensitization, phosphate, root response, tropism

## Abstract

Understanding how plants determine growth direction from environmental cues is important to reveal optimal strategies in plant survival. Nutritropism is the directional growth of plant roots towards nutrient sources. Our previous study showed that an NH_4_^+^ gradient stimulates nutritropism in the lateral roots, but not in the main roots, of a rice cultivar. In the present study, we report nutritropism in the main roots of rice accessions among the World Rice Core Collection, including WRC 25. We investigated the effects of components in nutrient sources on nutritropism in WRC 25. Nutritropism in main roots was stimulated by NH_4_^+^ and significantly enhanced by P*i*. We found that roots required more NH_4_^+^ stimulation for nutritropic responses in the presence of higher P*i*, meaning that P*i* desensitized root nutritropism. These results indicate that P*i* acts as an activator and a desensitizer in nutritropism. Such a regulation of nutritropism would be important for plants to decide their optimum growth directions towards nutrient sources, gravity, moisture, or other stimuli.

## 1. Introduction

Being immobile, plants use a wide range of physiological and morphological strategies to survive. Plants need to keep optimizing their responses to changing external conditions. To do so, they sense and respond to various environmental cues from biotic and abiotic sources, such as insects, pathogens, light, water and nutrients [1,2,3,4,5,6]. Recognition of what plants sense and how they respond has presented us with insights into their evolved strategies. The sensing of and responses to resources are complex, because essential resources, such as light, water, and nutrients, are harmful in excess. For example, to avoid photo-oxidative damage under high light intensity, plants rely on photorespiration [7]. Plants respond to excess water, which can cause hypoxia [8], and to excess nutrients in soil, which cause various adverse effects [9]. As these resources are essential for survival, they also need to regulate their shoot and root growth to acquire them efficiently [5,10,11,12]. Plants even recognize the directionality of environmental cues for these resources, and can reorient gradually through tropisms [13,14,15], and so grow towards or away from external stimuli [16]. Phototropism, gravitropism and hydrotropism have important roles in the acquisition of light and water. Tropism is defined as directional growth in response to a directional stimulus [14], and recently, we discovered a novel positive tropism towards nutrients, named nutritropism [17], in roots of a *japonica* rice (*Oryza sativa* L.) cultivar. Since primary functions of plant root system have been considered as the uptake of water and nutrients, and anchorage [18,19,20], nutritropism and the above three tropisms would be similarly important in terms of contributions to the root functions. Nutrients for plants distribute unevenly in environments [21], and nutritropism would be beneficial for plants to acquire nutrients efficiently. In the rice nutritropism we found that the stimulant is NH_4_^+^ gradient, which is a major chemical form of nitrogen nutrients available in natural environments. Plants show other local responses, such as lateral root branching, to locally available nutrients, including NH_4_^+^, in their root systems [22,23,24]. Thus, plants have developed to efficiently acquire nutrients, which distribute unevenly in environments. To ascertain how plants decide their growth direction from environmental cues and thus decide the optimal strategies for survival, a focus on tropisms is important. Interactions of environmental cues for root behaviors have been revealed in the characterization of tropisms. For example, root gravitropism interacts with hydrotropism [25,26,27]. Phytohormones, such as auxin, regulate tropistic bending [14,15], and the behavior of these hormones depending on the directions of external stimuli is a key factor to represent tropisms. In cases of auxin, PIN auxin transporters are polarly localized according to the direction of stimuli, and this asymmetric localization generates asymmetric auxin distribution [28]. Availabilities of some nutrients, including NH_4_^+^, also affect the auxin distributions in addition to the directional stimuli [29,30]. However, characteristics of nutritropism are poorly understood.

In a previous study, we found that an NH_4_^+^ gradient stimulates nutritropism in the lateral roots of Taichung 65, a *japonica* rice cultivar, but not in the main roots [17]. Here, we tested other rice germplasms for nutritropism in the main roots and characterized it. We screened accessions from the World Rice Core Collection (WRC) [31] and examined the effects of nutrient conditions on nutritropism in a bioassay system.

## 2. Results

### 2.1. Screening of Rice Accessions Showing Nutritropism in the Main Roots

To screen rice accessions showing nutritropism in their main roots, nutritropic bioassays using nutrient sources containing 200 mM NH_4_^+^, NO_3_^−^, K^+^, and P*i* were performed on the main roots of 50 accessions from WRCs. In this bioassay, within 24 h, while the main roots of some accessions always passed by the sources (Figure 1A), coiled responses at the sources were also observed in others (Figure 1B). This coiled response is directional towards the nutrient source. Therefore, it is reasonable to define this as a nutritropic response. We quantified the nutritropic responses of accessions from the frequencies of coiled responses among replications of the bioassays. Because the coiled response reflects an activity overcoming the stable gravitational effect, the frequency reflects the nutritropic activities against the stable forces toward gravity (gravitropism). Although the replications (*n* = 8–25) were not enough to accurately quantify their nutritropic responses, we found many accessions showing nutritropism in main roots (Appendix A). This indicated that the gradients of these mixtures were sufficient to stimulate nutritropism in their main roots. Using WRC 25, which had the second highest nutritropic response (38.1%), we further characterized nutritropism in main roots, because accessions with nutritropic responses that were too high, such as WRC 42 showing a 100% response, were likely improper to reveal the factors that increase the nutritropic response due to the almost maximum scores in our quantitative method.

### 2.2. Major Effects of NH_4_^+^ and PO_4_^−^ on Main Root Nutritropism

To assess the effects of components in the nutrient sources, we performed the nutritropic bioassay with various combinations of 200 mM NH_4_^+^, NO_3_^−^, K^+^, and P*i*. With all four nutrient components, WRC 25 showed a nutritropic response of 44.9% in many replications (*n* = 78, Figure 2). With NO_3_^−^, K^+^, and P*i* (no NH_4_^+^ condition), it showed no nutritropic response (*n* = 122). This result indicates that an NH_4_^+^ gradient is a stimulant for the nutritropism observed in main roots. However, WRC 25 showed a very low nutritropic response of only 3.51% towards NH_4_^+^ alone (*n* = 114). To assess whether such a low response was specific to WRC 25 or common among accessions, the top 11 accessions other than WRC 25 with high nutritropic responses (15.0–100%) (Appendix A) were assayed with NH_4_^+^ alone. All accessions showed little-to-no nutritropic response (0–15.4%, Appendix A), probably suggesting that a low nutritropic response towards a sole NH_4_^+^ source is a characteristic of nutritropism in main roots among the WRCs. These results also indicate that one or more of the other components—NO_3_^−^, K^+^, and P*i*—are important for nutritropism in addition to NH_4_^+^. Therefore, we assayed the main roots of WRC 25 with combinations of NH_4_^+^ and the other nutrients and compared these results to the responses toward NH_4_^+^ alone (Figure 2). The differences in frequencies were statistically analyzed by Fisher’s exact test. The roots showed similarly low nutritropic responses towards NH_4_^+^ + NO_3_^−^ (11.1%, *n* = 108, *p* = 0.297) and NH_4_^+^ + NO_3_^−^ + K^+^ (4.88%, *n* = 123, *p* = 1.000), compared to NH_4_^+^ alone. In contrast, they showed a significantly higher response of 33.9% toward NH_4_^+^ + P*i* (*n* = 127, *p* < 0.001). This response was comparable to that of all four components, 44.9% (*p* = 0.277). These results indicate that P*i* is another key nutrient for nutritropism in rice main roots. Similarly, significantly higher nutritropic responses were observed with nutrient sources, including NH_4_^+^ + P*i* + K^+^ (31.6%, *n* = 117 *p* < 0.001) and NH_4_^+^ + P*i* + NO_3_^−^ (26.1%, *n* = 188, *p* < 0.001), compared to NH_4_^+^ alone. In comparison to the results of NH_4_^+^ + P*i* with those of NH_4_^+^ + P*i* + K^+^ and NH_4_^+^ + P*i* + NO_3_^−^, there is no significant difference (*p* = 1.000, *p* = 0.693, respectively).

### 2.3. Activation of Main Root Nutritropism by Non-Gradient Pi

To elucidate whether the activation of nutritropism in response to P*i* requires its gradient, the nutritropic responses of WRC 25 were examined in uniformly increasing concentrations of P*i* in the agar medium. The nutrient source contained 200 mM NH_4_^+^ alone. The nutritropic responses were 4.63% in the medium containing 10 mM P*i* (*n* = 108) and 14.4% in 15 mM P*i* (*n* = 118) (Figure 3). Relative to that without supplementary P*i*, nutritropic responses did not increase at 10 mM (*p* = 0.743), but increased significantly at 15 mM (*p* < 0.01). These results show that non-gradient P*i* is sufficient to activate nutritropism toward NH_4_^+^. Since we prepared the P*i* with an NaOH-neutralized NaH_2_PO_4_ solution (see Materials and Methods), P*i* supplementation substantially increased the Na^+^ concentration (for example, 100 mM P*i* condition contained 154 mM Na^+^). It was possible that Na^+^ affected our conclusion. Therefore, to investigate the effects of Na^+^ on our observations, we used NaCl for the Na^+^ source and evaluated the nutritropic responses toward the Na^+^ and NH_4_^+^ conditions. The same concentrations were set as those at the two peaks in Figure 4, but with no P*i*. The responses were 5.33% (*n* = 75) toward 50 mM NH_4_^+^ + 154 mM Na^+^ and 5.06% (*n* = 79) toward 150 mM NH_4_^+^ + 461 mM Na^+^, not significantly different from those toward NH_4_^+^ alone (*p* = 0.072 and 0.370, respectively; Appendix A). Therefore, the effect of Na^+^ on our observation was negligible.

### 2.4. Pi Desensitization of Nutritropism

Our results demonstrate that the nutritropic response in rice main roots mainly depends on the availability of both NH_4_^+^ and P*i*. To further investigate the dependence, we evaluated nutritropic responses to a broad range of NH_4_^+^ and P*i* concentrations from 25 mM each to 250 mM each, where NH_4_^+^ and P*i* concentrations were always the same, and drew an NH_4_^+^ + P*i* dose–response curve (*n* = 61–84, Appendix A). The curve seemed to have peaks at around 50 mM and 175–200 mM NH_4_^+^ + P*i*. According to a scenario of desensitization (see discussion), NH_4_^+^, P*i* or both might affect nutritropism sensitivity, but NH_4_^+^ alone caused little nutritropism in the first place. Since P*i* greatly changed the nutritropic response towards NH_4_^+^ (Figure 2), we hypothesized that P*i* impacts the sensitivity of nutritropism towards NH_4_^+^.

If our hypothesis is true, the NH_4_^+^ dose–response curve should shift along the NH_4_^+^ axis in response to P*i*. Using nutrient sources containing 10 to 250 mM NH_4_^+^, we first obtained an NH_4_^+^ dose–response curve (*n* = 77–114, Figure 4, blue line). The nutritropic response was low (<13.0%) at all NH_4_^+^ concentrations, seemingly with one small peak at 50 mM NH_4_^+^. In the presence of 100 mM P*i* in the NH_4_^+^ nutrient sources, the nutritropic responses were dramatically increased (*n* = 82–168, Figure 4, red line): the curve shifted upwards, with a large peak at 50 mM NH_4_^+^. This shift means that 100 mM P*i* activated nutritropism, but does not mean that P*i* changed the sensitivity. Relative to the curve at 100 mM P*i*, however, those at 200 mM (*n* = 71–127) and 300 mM (*n* = 88–181) P*i* showed a shift to a higher amount of NH_4_^+^ (rightwards; Figure 4, green and purple lines, respectively). Especially in the presence of 300 mM P*i*, there was a significant peak at 150 mM NH_4_^+^, as high as that at 50 mM NH_4_^+^ in the presence of 100 mM P*i*. Thus, the NH_4_^+^ dose–response curve for the nutritropic response shifted to a higher NH_4_^+^ owing to the increased P*i* availability. In other words, nutritropism required a greater stimulation of NH_4_^+^ at higher levels of available P*i* to produce the same nutritropic response as lower levels of available P*i*. Comparing the nutritropic responses at 50 mM NH_4_^+^, nutritropic responses in the presence of 100 mM P*i* (54.1%, *n* = 137) was significantly higher than in 0 mM (13.0%, *n* = 77, *p* < 0.001), 200 mM (29.6%, *n* = 71, *p* < 0.01), and 300 mM P*i* (27.0%, *n* = 111, *p* < 0.001). In the cases of the results at 150 mM NH_4_^+^, the nutritropic response in the presence of 300 mM P*i* (57.5%, *n* = 181) was significantly higher than in 0 (1.25%, *n* = 80, *p* < 0.001), 100 mM (32.7%, *n* = 168, *p* < 0.001), and 200 mM P*i* (42.2%, *n* = 83, *p* < 0.05).

### 2.5. Nutritropic Response in WRC 1

The P*i*-dependent activation and desensitization of nutritropism suggested that the interaction of NH_4_^+^ and P*i* should be considered to investigate nutritropic responses. The variety of nutritropic responses shown in Appendix A represents the combined responses of each accession to 200 mM NH_4_^+^ and P*i*, but it remains to be confirmed whether the result shows a variety of preferences in nutritropism among the WRCs. For example, one of the most major accessions in the study of rice WRC 1 (Nipponbare), which showed almost no nutritropism (only 1 coiled response in 25 bioassays), might have a high nutritropic response at different conditions of NH_4_^+^ and P*i*. To investigate the nutritropic responses of WRC 1 toward various combinations of NH_4_^+^ and P*i*, we assayed its main roots with nutrient sources containing 10, 50, 100, 150, 200 or 250 mM NH_4_^+^ and 100 or 200 mM P*i*. Although WRC 25 showed great differences in its nutritropic response to these combinations (Figure 4), WRC 1 always showed low nutritropic responses towards NH_4_^+^ (0.00–1.49%, *n* = 67–93) at both P*i* concentrations (Appendix A).

## 3. Discussion

In this study, we found that 21 accessions showed coiled responses towards nutrient sources (Appendix A). Although the concentrations of all the nutrients in our nutrient sources were relatively high, root growth did not stop around the sources (Figure 1). This coiled response was directional towards nutrient sources and, therefore, it is reasonable to define coiled response as a nutritropic response. We further found that the stimulus for this directional response was NH_4_^+^ in the sources, at least in WRC 25 responses (Figure 2). It is known that NH_4_^+^ inhibits gravitropism [30] but does not reverse it, so the inhibition of gravitropism cannot explain the coiled response. A previous report showed that an NH_4_^+^-induced low pH in roots triggered a helical response of the rice root [32]. It is not reasonable to consider this helical response, which was shown in the previous study, as a part of nutritropism. The authors caused such helical responses with uniform NH_4_^+^ treatments in a hydroponic culture, meaning that the helical responses did not depend on the direction of NH_4_^+^ stimulation. In fact, they observed an asymmetric distribution of auxin in rice root tips in response to uniform NH_4_^+^ treatments, and there was only one directional factor, gravity, in their experimental conditions. Therefore, it is very difficult to determine that uniform NH_4_^+^ treatments directly cause the asymmetric distribution of auxin, but reasonable to interpret that the helical response resulted from the disruption of root gravitropism, as the authors concluded. This conclusion is consistent to the NH_4_^+^-mediated inhibition of gravitropism shown by Zou et al. [30]. If these phenomena were reproduced in our experimental conditions, the coiling directions should be random, but not always toward the nutrient sources. Thus, nutritropism toward NH_4_^+^ was a distinct behavior, compared to the previous studies.

We selected WRC 25 to investigate the effects of other nutrients on the response for its moderate frequency of nutritropism among our tested accessions. NH_4_^+^ is the stimulus for nutritropism in lateral roots, as observed in Taichung 65 [17]. This is why we first examined the NH_4_^+^ effect among the NH_4_^+^, NO_3_^−^, K^+^, and P*i*. Because the stimulation of nutritropism in both roots is an NH_4_^+^ gradient, the molecular mechanisms of nutritropism in main and lateral roots would be similar.

P*i* was another key factor in nutritropism. It dramatically activated nutritropism towards NH_4_^+^ (Figure 2 and Figure 4). This does not suggest that the activation of nutritropism is an additive effect of two tropistic stimulations by NH_4_^+^ and P*i*, because P*i* was not a nutritropic stimulant as follows: P*i* in the nutrient source never caused a coiled response without NH_4_^+^ (Figure 2) and non-gradient P*i* activated nutritropism (Figure 3). Because this nutritropic character seems to be shared among WRCs (Appendix A), P*i* activation of nutritropism towards NH_4_^+^ would be a reasonable survival strategy for rice plants. N and *p* are the principal nutrients that limit plant productivity [6] and are deficient in general soil environments. Therefore, we believe that N and *p* distribution in soil is sometimes patchy, causing nutrient hot spots around root systems. Our results indicate that rice plants do not seek nutrient hot spots only containing NH_4_^+^ or P*i*, but seek both (such as those deposited by livestock). The nutrient sources in this study contain several µmol of nutrient molecules. According to a previous study, 1 kg of animal fresh manure contains 3.6 g NH_4_-N and 0.7 g *p*, respectively [33]. Therefore, it is possible that the nutrient hotspots we applied in our bioassay systems are reproduced in natural soil environments only with 100 mg of fresh manure.

Another important character of nutritropism that we revealed is its desensitization. Biological systems that sense external stimuli include desensitization mechanisms, which enable the sensory systems to respond to a broad range of stimulation intensities [34]. Otherwise, strong stimulation would saturate the sensory systems and inhibit their ability to recognize the direction of stimuli. Additionally, in plant phototropism, desensitization is important in responses to a broad range of light intensities. In its presence, increasing light intensity causes two peaks of phototropic response [35,36]. In our results, the two-peak shape of the NH_4_^+^ + P*i* dose–response curve (Appendix A) was strikingly similar to the phototropic dose–response curve in plant shoot tips. Our study shows that plant roots have systems for desensitization to external stimuli as well as shoot phototropism. Interestingly, nutritropism desensitization was triggered by the non-stimulant P*i* (Figure 4). Thus, P*i* has a significant role in nutritropism as an activator and a desensitizer in roots, at least in WRC 25 rice accession.

Focusing on the main root responses, our study indicates the following characteristics of nutritropism (Figure 5): the nutritropic sensing of an NH_4_^+^ gradient becomes sensitive at a moderate P*i* availability and insensitive at a high P*i* availability (Figure 4). These characteristics enable plant roots to continue to accurately recognize the direction of nutrient hot spots containing NH_4_^+^ and P*i* during the approach of root tips to the sources. In addition, at low P*i* availability, rice roots do not grow towards NH_4_^+^ sources (Figure 2 and Figure 4). Plants sometimes need to direct their root growth gravitationally to acquire water and to anchor their shoots (or in other directions for other purposes) rather than nutrients, even in the presence of nutritropic stimulation. Our revealed regulation of nutritropism would be important for plants to decide their optimal growth direction towards resources. It is well known that different tropisms, such as gravitropism and hydrotropism, interact [25,26,27], and nutritropism should also interact. Our findings offer a new perspective for interpreting the effects of nutrients on root tropisms. Interestingly, it is well known that NH_4_^+^ and P*i* affect gravitropism, too [29,30]. The inhibition of gravitropism by high NH_4_^+^ or by low P*i* levels might represent an interaction between gravitropism and nutritropism, although the latter inhibition of gravitropism by low P*i* levels, may be inconsistent with the enhancement of nutritropism by a high P*i* level in our study. It is of interest to explore how plant roots organize or optimize their growth directions under various tropistic stimulations.

Of further interest are the molecular mechanisms. As described above, it is well known that NH_4_^+^ inhibits gravitropism, probably due to NH_4_^+^-induced low pH [30,32]. This cannot explain nutritropic bending toward NH_4_^+^, even if gradient NH_4_^+^ induced a lower pH status in the stimulating sides of root tips. This is because an asymmetric pH status generally promotes differential cell elongation, resulting in bending toward a high pH side [37], and this direction is opposite to the nutrient sources. Therefore, mechanisms of tropistic responses toward an NH_4_^+^ source are inexplicable from this prior knowledge, although an NH_4_^+^-mediated inhibition of gravitropism seems to be a mechanism of gravitropism and nutritropism interaction. Interestingly, the availability of NH_4_^+^ is strongly related with auxin signaling in roots, as described above. A recent study reported that NH_4_^+^ deficiency increases auxin signaling in roots [38]. Unknown systems involved in NH_4_^+^ sensing, such as NH_4_^+^ receptors, might be expressed in roots, and the signal would be transduced to auxin signaling. Nutritropism should also require sensing mechanisms for the NH_4_^+^ gradient, and this signal might be transduced to auxin signaling. Laser ablation is a promising technique to investigate the tissues that are responsible for the sensing of the NH_4_^+^ gradient, as is the case of gravitropism research [39]. Accessions or mutants with different responses of nutritropism allow us to dissect them. WRC 1 showed very low nutritropic responses (Appendix A). Therefore, WRC 1 and WRC 25 are promising resources for investigating the genetics of nutritropism. The accessions not showing nutritropism preferred gravitropism. Such a preference in soil environments would increase their water availability via deep rooting. A variety of nutritropic responses among accessions or species presents us with an insight into the significance of nutritropism for plant survival and evolution. It might be possible that some plant roots can show nutritropism toward nutrients other than NH_4_^+^.

## 4. Materials and Methods

### 4.1. Plant Materials and Growth Conditions

We tested 50 accessions from the WRC (Appendix A). Sterilized seeds were germinated and grown on square plates (sterile square Schale No. 2, Eiken Chemical Co., Ltd., Tokyo, Japan) held at 60° from the horizontal position in a growth chamber under continuous light at 28 °C. Each plate contained 60 mL of 1/200 diluted MS medium (pH 5.7–5.8; Murashige and Skoog plant salt mixture, Wako Pure Chemical Industries Ltd., Osaka, Japan) supplemented with 2% (*w/v*) sucrose (guaranteed reagent; Fujifilm Wako Pure Chemical Corporation, Osaka, Japan), 0.05% (*v/v*) Plant Preservative Mixture (Plant Cell Technology Inc., Washington, DC, USA), and 1.5% (*w/v*) agar (Agar Purified Powder, Nacalai Tesque, Kyoto, Japan). When P*i*-supplemented 1/200 diluted MS media was prepared, a 2 M P*i* solution was added to increase the P*i* by 10 and 15 mM.

### 4.2. Preparation of Nutrient Sources for the Nutritropic Bioassay

Almost all the procedures used to prepare the nutrient sources were the same as in Yamazaki et al. [15], except that we used 1.5% (*w/v*) agar for the media. Two M stock solutions of NH_4_Cl, NH_4_NO_3_, KNO_3_, KCl, or NaH_2_PO_4_·2H_2_O (Wako Pure Chemical Industries Ltd.) were used as nutrient sources in different combinations at 0–250 mM. In the case of the NaH_2_PO_4_·2H_2_O solution, NaOH was added at 1.07 M to neutralize it (pH = 7.4).

### 4.3. Nutritropic Bioassay

The nutrient sources to generate the nutrient gradients were inserted vertically with the tip reaching the bottom of the plate 3 to 5 mm from the main root tips (primary or crown roots) of 3- to 10-day-old seedlings in the direction of root elongation (Figure 1). The root responses were defined as the following: the root growth that passed by a source was recorded as ‘passed’ (Figure 1A), and the root growth coiling around a source more than once was recorded as ‘coiled’ (Figure 1B). The former was elongation in the gravitational direction, as observed in the main roots of Taichung 65 [17]. The latter was a nutritropic response, because the growth was in the direction toward the nutrient source.

### 4.4. Statistical Analyses

Statistical significance of the differences in the proportions of the root responses (‘passed’ or ‘coiled’) in different conditions were tested using Fisher’s exact test [40]. In cases of multiple comparisons, *p*-values were corrected by the Holm method [41]. These tests were undertaken using the ‘RVAideMemoire’ package (v. 0.9–75) for R statistical software [42] at *p* < 0.05 in all tests.

## Figures and Tables

**Figure 1 plants-11-00733-f001:**
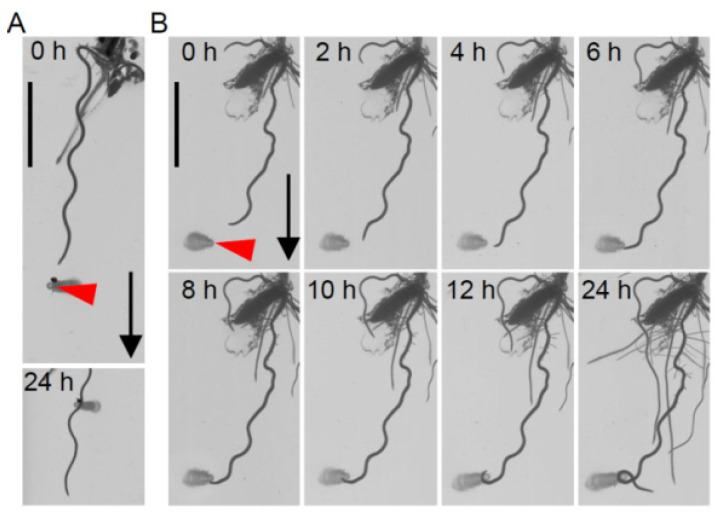
Nutritropic bioassay in rice main roots. The images show examples of nutritropic bioassay for the qualitative evaluation of nutritropic responses in main roots. Main roots of rice seedlings cultivated in culture agar media were assayed with nutrient sources containing 200 mM of NH_4_^+^, NO_3_^+^, K^+^, and/or P*i*. Nutrient sources (red arrowheads) were inserted in the direction of the elongation vector at the 3–5 mm distance from the main root tips. Growth responses of the main roots after 24 h were defined as the following two behaviors; (**A**) passed by or (**B**) coiled more than once at the inserted nutrient sources. WRC1 and WRC25 were used as examples in (**A**,**B**), respectively. Both accessions were cultivated in 1/200 diluted MS medium (pH 5.7–5.8) supplemented with 2% (*w/v*) sucrose, 0.05% (*v/v*) Plant Preservative Mixture, and 1.5% (*w/v*) agar. Black arrow, gravitational direction; scale bars, 10 mm.

**Figure 2 plants-11-00733-f002:**
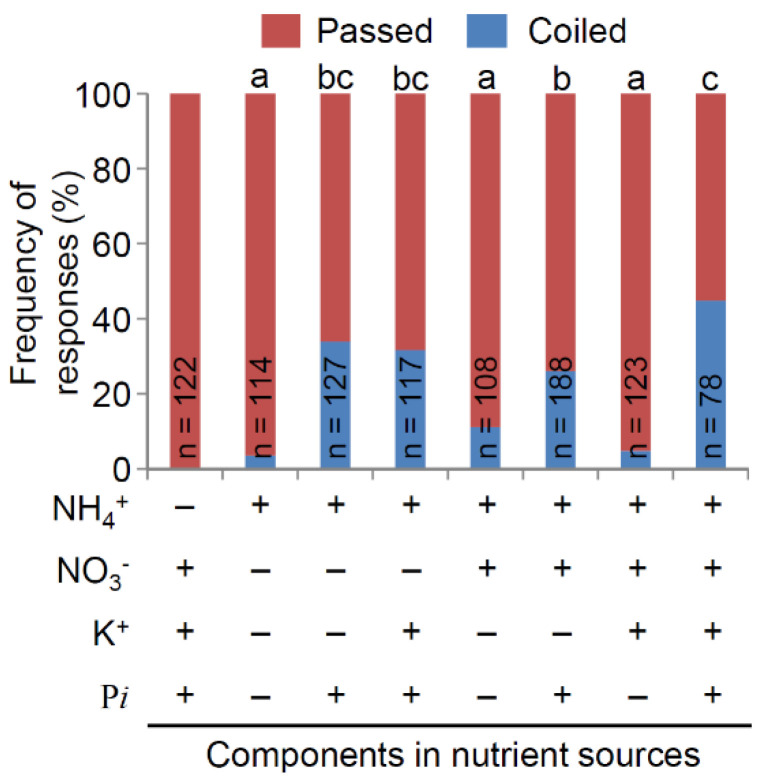
Nutritropic responses of the main root (passed or coiled) of WRC 25 to the components of the nutrient sources. The frequencies of passed and coiled responses were determined in the nutritropic bioassay with nutrient sources containing various nutrient combinations of NH_4_^+^, NO_3_^−^, K^+^, and/or P*i* at 200 mM. Statistical significance was tested using Fisher’s exact test with multiple testing corrections (Holm correction). Different letters on the bars indicate a significant difference at *p* < 0.05.

**Figure 3 plants-11-00733-f003:**
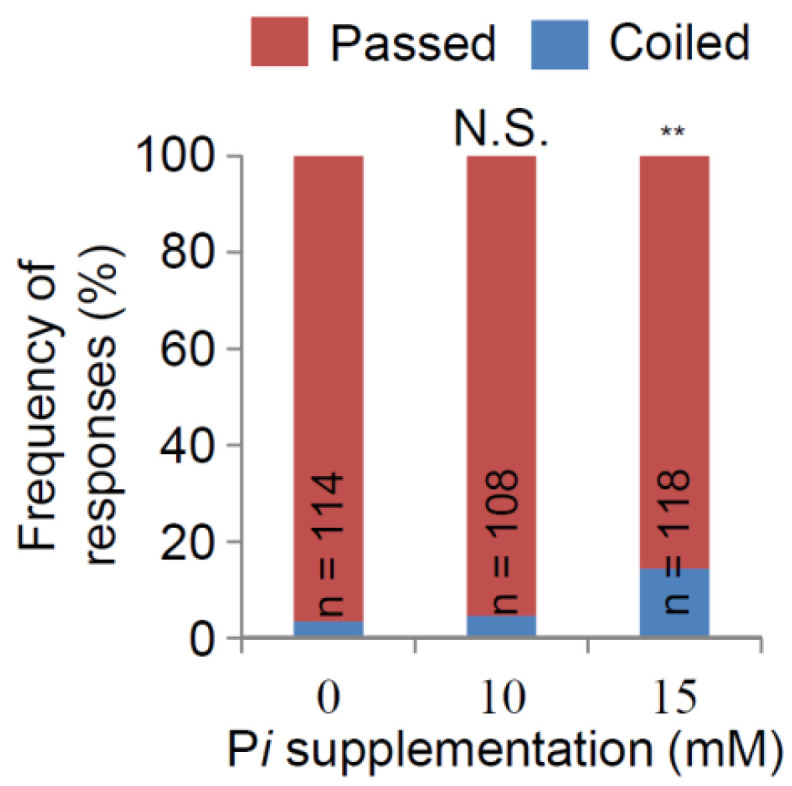
Increase in the nutritropic response of the main root (passed or coiled) of WRC 25 to NH_4_^+^ in the P*i*-supplemented culture medium. Frequencies of the passed and coiled responses were determined in the nutritropic bioassay with a 200 mM NH_4_^+^ nutrient source. Data of 0 mM P*i* were the same as those in Figure 2. The statistical significance was tested using Fisher’s exact test. N.S., not significant; **, *p* < 0.01.

**Figure 4 plants-11-00733-f004:**
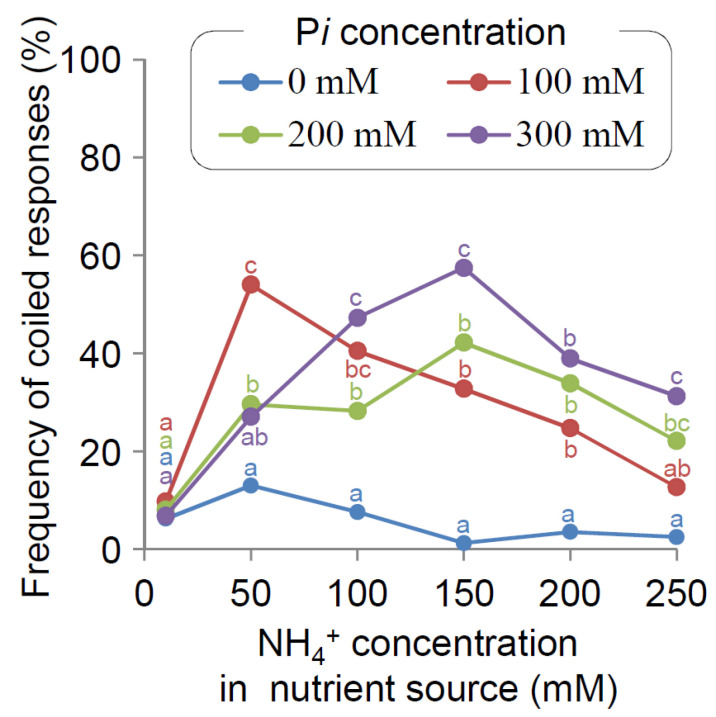
Effect of the interaction of NH_4_^+^ and P*i* on the nutritropic response of the main roots of WRC 25. Frequencies of the coiled responses in the main roots of WRC 25 were determined in the nutritropic bioassays (*n* = 71–181 per each condition) with nutrient sources containing NH_4_^+^ and P*i* at the indicated concentrations. Data of 200 mM NH_4_^+^ and 200 mM P*i* were the same as those in Figure 2. Statistical significance among conditions at the same concentrations of NH_4_^+^ was tested using Fisher’s exact test with multiple testing corrections (Holm correction). Different letters indicate a significant difference at *p* < 0.05.

**Figure 5 plants-11-00733-f005:**
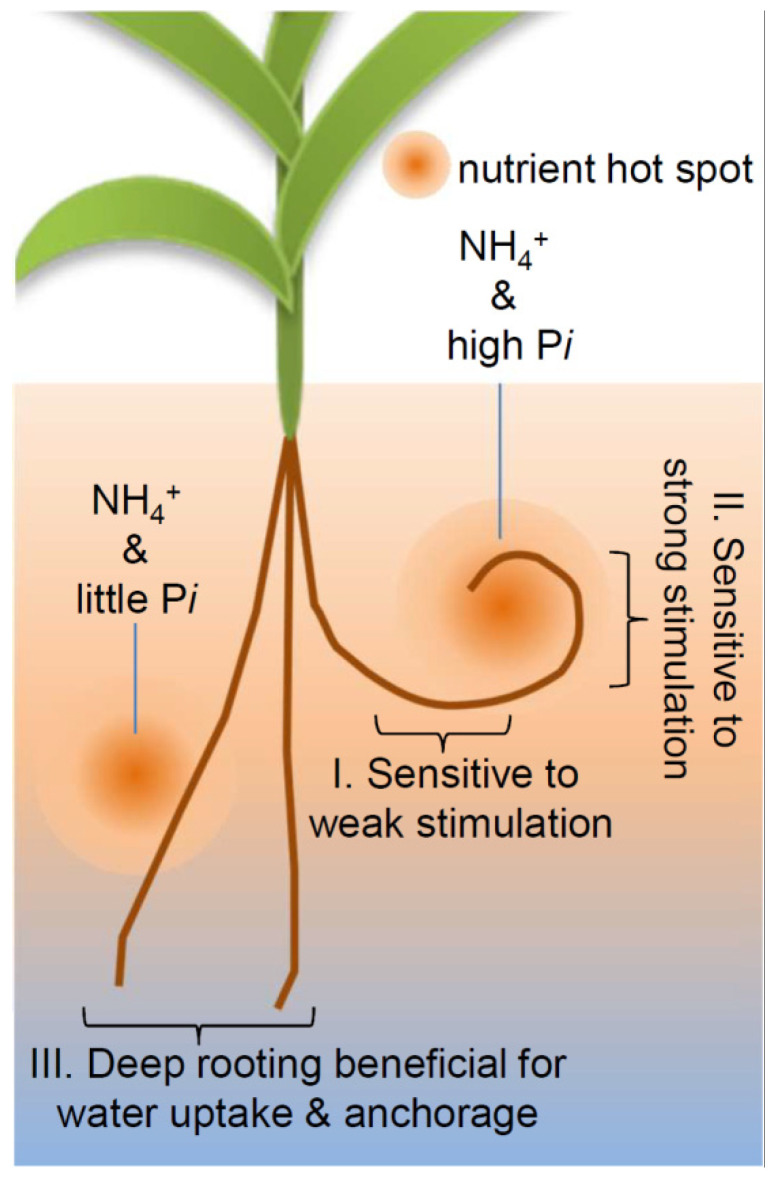
A hypothetical model of tropistic responses in a natural environment. I: when the root tips are exposed to nutrient gradients from a hot spot of NH_4_^+^ and high P*i*, they can respond to weak stimulation by NH_4_^+^ at a distance from the hot spot. II: as the root tips approach the hot spot, nutrient intensities become stronger. Increasing P*i* enables root tips to respond to the increasing intensities. III: when a hot spot contains NH_4_^+^ but not P*i*, roots pass by them. In general, deep rooting through gravitropism enables plants to anchor themselves and to acquire water efficiently. Nutrient hot spots can be provided by livestock.

## Data Availability

The data that support the findings of this study are available from the corresponding author upon reasonable request.

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
