# Peer review of "The Effect of Phosphate on the Activity and Sensitivity of Nutritropism toward Ammonium in Rice Roots"

_plants, 2022, doi:10.3390/plants11060733_

Round 1

Reviewer 1 Report

The manuscript reported some data regarding the nutritropism phenomenon. The analysis was performed using different rice accessions, that permitted to highlight that the responses is strictly depending by peculiar genetic background. At the same time, the paper presents only data regarding the responses of root growth. This permit only a preliminary description of the phenomenon. It could be also underlined that a first characterization of nutritropism phenomena in rice has been already reported in a previous article published by the same authors. In other words, the originality of the present article is quite low. Moreover, the paper contains some sentences that are not clear or too speculative as well as there are some formal mistakes that must be removed.

Comments and Suggestions for Authors

  • The authors used PO43- to indicate inorganic phosphate. Considering that the experiments was performed using a pH near to neutral values, the two forms present are HPO42- and H2PO4-. From the formal point of view to use PO43- is not correct. Please change PO43- in Pi throughout the text.

Introduction

  • Line 36: the reference 13 could be important from the historical point of view, but other more recent references must be added.
  • Lines -38-39: The authors produced only preliminary evidence about a new possible tropism. Please reformulate this sentence considering that further work is requested to sustain this conclusion. At the moment there are not any evidence at the molecular and physiological level that permit to compare adequately this phenomenon with other tropisms, that are very well characterized. This aspect must be considered in all parts of the manuscript.
  • Other aspect of introduction: The authors did not introduce some crucial aspects that are strictly linked to the mineral nutrient acquisition, that should be considered to clarify the real role/significance of nutritropism. For example, the movement of the nutrient from soil to root (i.e. interception, mass flow and diffusion contributes), the rhizospheric environment, peculiar traits of different species). All these aspects deeply influence nutrient acquisition by plants. What relationships they might have with this new tropism? The evidence available until now are being obtained studying the root growth in a agar system, that is a useful experimental system, but that is a very simplified system, compared to the soil environment. The authors must consider this aspect as well as to stress that a very lot of work is requested before to equate nutritropism with other tropisms. This last comment regards not only the introduction, but also the discussion in which some sentences are very too speculative.

Results

  • Paragraphs 2.4 and 2.5: in both these paragraphs the authors to discuss the results. Considering the role of this section, these parts must be moved in the discussion one.

Discussion

  • The results reported in this paper highlighted that many of the genotypes tested did not show the nutritropism response. This is a crucial aspect that must be considered.

The authors debated in many points of the discussion on the interaction that could exist between nutritropism and gravitropism. This last tropism is ubiquitarian in quite all species (obviously without considering the mutants), whilst the half of the accessions that are tested do not show nutritropism. A comment regarding this point is requested.

  • The authors did not consider that in many of the experiments were used a very high concentration of nutrients, For example: the experiment reported in figure 4 evaluate concentrations very higher of Pi and ammonium (300 mM and 250 mM, respectively). This aspect must be discussed.
  • Lines 214-216: this sentence is not clear.
  • Lines 244-255: the authors proposed a parallelism between the desensitization induced by high light intensity with those induced by high Pi concentrations. This conclusion is very questionable. In order to support it, the authors must produce new data that can permit to exclude the possible toxic effect induced by these Pi concentrations (osmotic effect? others? etc.)
  • Lines 276 - : the authors speculated on the possible action mechanism of ammonium to affect gravitropism. They assume that this cation could counteract the pH changes involved in the transduction of the gravitropism signal. The authors did not have any experimental evidence as well as did not consider the different responses induced by this nitrogen form (i.e. effect on electric potential of PM, nitrogen metabolism etc. etc.). Without experimental evidences to support this hypothesis, all this part must be removed.

Other parts of the discussion require a very deep revision ……

Author Response

Response to Reviewer 1 Comments

Point 1: The authors used PO43- to indicate inorganic phosphate. Considering that the experiments was performed using a pH near to neutral values, the two forms present are HPO42- and H2PO4-. From the formal point of view to use PO is not correct. Please change PO43- in Pi throughout the text.

Response 1: Thank you for your important comment. Our description was wrong, and we have changed all PO43- in Pi throughout the text and Figures.

Point 2: Line 36: the reference 13 could be important from the historical point of view, but other more recent references must be added.

Response 2: Thank you for your kind suggestion. We put recent two references published in 2014 and 2020.

Point 3: Lines -38-39: The authors produced only preliminary evidence about a new possible tropism. Please reformulate this sentence considering that further work is requested to sustain this conclusion. At the moment there are not any evidence at the molecular and physiological level that permit to compare adequately this phenomenon with other tropisms, that are very well characterized. This aspect must be considered in all parts of the manuscript.

Response 3: Thank you for your pointing out that our statement could be understood in this way. We understood your meanings, and we think that evidences at the molecular and physiological level are very important to compare mechanisms in tropisms. When we compare the responses of tropisms, molecular evidences will be important to consider interaction among tropisms. Another important thing is that tropisms are defiened as “directional growth toward a directional stimualus”. Actually, well-knonw tropisms such as phototropism, gravitropism, and hydrotropism have been accepted as distinct and important tropisms before showing evidences at the molecular and physiological level at the moments, they were found. This is because the direction of each tropic bending is independent each other. In our manuscript, we basically have described the characteristics, aspects of nutritropic resposnes, and therefore, carefully described the nutritropic direction is independent from gravitational direction. We added the definition of tropisms with the refference at Line 38, “Tropism is defined as directional growth in response to a directional stimulus [14].”.

Point 4: Other aspect of introduction: The authors did not introduce some crucial aspects that are strictly linked to the mineral nutrient acquisition, that should be considered to clarify the real role/significance of nutritropism. For example, the movement of the nutrient from soil to root (i.e. interception, mass flow and diffusion contributes), the rhizospheric environment, peculiar traits of different species). All these aspects deeply influence nutrient acquisition by plants. What relationships they might have with this new tropism?

Response 4: Thank you for your brilliant suggestions. Our manuscipt lacked this view point you raised, and we should include the distributions of the stimulus, NH4+ at least, to increase significaces of tropisms toward nutrients. As you know, nitrogen is a principal nutrient that limits plant growth in natural environments, and our study will be attracttive by this improvement. On the other hand, in this paper we prefer to focus on the growth response aspects. So we will leave the aspects, which are mixed with general characteristics of nuteirnt uptake by roots and nutritropism, to other research. We have added information of nutrient distribution and responses of root systems to locally available nutrients in the revised manuscript.

Point 5: The evidence available until now are being obtained studying the root growth in a agar system, that is a useful experimental system, but that is a very simplified system, compared to the soil environment. The authors must consider this aspect as well as to stress that a very lot of work is requested before to equate nutritropism with other tropisms. This last comment regards not only the introduction, but also the discussion in which some sentences are very too speculative.

Response 5: Thank you for your sharp opinion. It is belived that gravitropism/hydrotropism (and negative phototropism) in roots contribute to acquitions of water and anchorage. We meant that nutropism would be similarly important, because primary root functions are to acquir water/nutrients and to anchor their shoots. We have clarified this explanation in the introduction, “Since primary functions of plant root system have been considered as uptake of water and nutrients, and anchorage [18-20], nutritropism and the above three tropisms would be similarly important in terms of contributions to the root functions.”.

In another aspect you raised, many studies on tropisms (or other root responces) have been examined in agar or non-soil experimental systems. These conditions are muh different from soil or natural environments, and we should carefully consider the results. Whereas we think that your pointing is right, authors of such studies have concluded the contributions of tropiosms in natural environments. We do not know reports to show the actual contributions of tropisms in the natural environments exept a Dro1 paper reported by Haga et al. in 2020. Therefore, adding the sentences “Since primary functions~”, we hope that you allow us to compare nutritropism with the other tropisms.

Point 6: Paragraphs 2.4 and 2.5: in both these paragraphs the authors to discuss the results. Considering the role of this section, these parts must be moved in the discussion one.

Response 6: Thank you for your suggestion. We have moved the sentenses in paragraph 2.4 in the discussion section. Owing to your kind suggestion, the repeated descriptions have become single. In the beginning of paragraph 2.5, we need to explain our motivation to investigate nutritropism of WRC 1 in detail. The last sentence was deleted, because a similar sentense are present in the Discussion section.

Point 7: The authors debated in many points of the discussion on the interaction that could exist between nutritropism and gravitropism. This last tropism is ubiquitarian in quite all species (obviously without considering the mutants), whilst the half of the accessions that are tested do not show nutritropism. A comment regarding this point is requested.

Response 7: Thank you for your very interesting suggestion. We suppose that growth toward water would be beneficial for plant survival more than gwowth toward nutrients in general, because increasing of water availability also increases availabilty of nutrients dissolved in the water. Although water distribution is uneven in soil environments, water availability generally inceases with soil depth. So, we consider that gravitropism, which cause deep rooting, is a dominant tropism for root functions and dominant in many plant species. Our rsults meant that the accessions not showing nutritropism prefered gravitropism. We added sentences in discussion section, “The accessions not showing nutritropism preferred gravitropism. Such preference in soil environments would increase their water availability via deep rooting.”, before the original sentense “Variety of nutritropic response among accessions or species will give us insights into the significance of nutritropism for plant survival and evolution.”.

Point 8: The authors did not consider that in many of the experiments were used a very high concentration of nutrients, For example: the experiment reported in figure 4 evaluate concentrations very higher of Pi and ammonium (300 mM and 250 mM, respectively). This aspect must be discussed.

Response 8: Thank you for your kind consideration. As you mentioned, we agree that our using concentrations were much high. As far as we know, there is no report to apply nutrients at the pinpoints around only a small part of a root, and we honestly think that comarison of conditions between our study and previous reports, using uniform applicarions, is very difficult. We can only say that the root growth did not stopped in our experimental conditions (e.g. Pi, ammonium, K+ and NO3- at 200 mM as shown in Figure 1), although the roots coiling at the nutrient sources wolud be exposed at relatively high concentrations. We have now acknowledged this point and added a sentense in the Discussion section of the revised manuscript, “Although concentrations of all nutrients in our nutrient sources were relatively high, root growth did not stop around the sources (Figure 1).”.

Point 9: Lines 214-216: this sentence is not clear.

Response 9: Thank you for your pointing out. As you felt, we now think this should be more simplified. We have modified this to “The authors caused such helical responses with uniform NH4+ treatments in hydroponic culture, meaning that the helical responses did not depend on the direction of NH4+ stimulation”.

Point 10: Lines 244-255: the authors proposed a parallelism between the desensitization induced by high light intensity with those induced by high Pi concentrations. This conclusion is very questionable. In order to support it, the authors must produce new data that can permit to exclude the possible toxic effect induced by these Pi concentrations (osmotic effect? others? etc.)

Response 10: Thank you for your kind consideration and sorry for you confusion. We did not propose such parallelism but wanted to explain our considerations why we focused on desensitization. We would like to think desensitization as a characteristic of a response, as previous authors have shown in their reports on phototropic desensitization. They have also concluded the desensitization characteristic in phototropism only from the responses toward light with various intensities. You may be claiming that the molecular mechanisms or reasons should be clarified to say desensitization, probably because our description was very confusing. We do not think that you feel questionable about phototropic desensitization even if toxic stress by high light, such as excess H2O2, causes signals to represent the desensitization.

In order to avoid potential confusions, we have deleted the sentence, “The molecular basis for this desensitization has been revealed [24,27], and high light intensity makes photoreceptors insensitive via interaction with other components.”.

Point 11 Lines 276 - : the authors speculated on the possible action mechanism of ammonium to affect gravitropism. They assume that this cation could counteract the pH changes involved in the transduction of the gravitropism signal. The authors did not have any experimental evidence as well as did not consider the different responses induced by this nitrogen form (i.e. effect on electric potential of PM, nitrogen metabolism etc. etc.). Without experimental evidences to support this hypothesis, all this part must be removed.

Response 11: Thank you for your deep consideration and really sorry for your confusion. We completely agree with you. In fact, we do not speculate them, but Jia et al. did so in their previous paper. We discussed the relationship between nutrirtopism and these previous findings because the two referees requested us to do so, when we submitted this manuscript to a different journal last year.

Here, we would like to describe that the previous findings, helical growth in response to low pH induced by ammonium, cannot explain nutritropism, although it was possible that ammonium gradient changed pH in our experiments. We have now modified these sentences to avoid any potential confusion: “~ induced low pH [30,32]. This cannot explain nutritropic bending toward NH4+, even if gradient-NH4+ induced lower pH status in the stimulating sides of root tips. This is because asymmetric pH status generally promotes differential cell elongation, resulting in bending toward high pH side [29], and this direction is opposite to the nutrient sources.”

Reviewer 2 Report

The manuscript submitted by the authors has put novel information on newly introduced tropism as reported by the same research group earlier. Here, in this manuscript, they show the nutritropism in the main roots of rice accessions among the World Rice Core Collection. The results described conclude a story that indicates that PO4 3− acts as an activator and a desensitizer in nutritropism. It is an interesting study though, however still more information should be supported in this regard that this coiling response arises in such roots on actual nutritropism, I have some general questions to authors as suggested below:

Firstly, the introduction part is very short in the manuscript, maybe talk about auxin transport/gradient in root bending and development.
Do authors have an idea about the gene expression of various auxin-related genes like the ones specific to auxin transporters in the coiled region? Is it an auxin-based response or other hormonal networking? What about PIN transporters and their localization during such coiling?

Do authors define if it is the tip of the root or the cells adjoining the tip or above that sense the nutritropic behavior? What senses the behavior in cells? Is this behavior independent of starch sedimentation? That should be then checked by iodine staining. The bigger question here still stays then what defines or senses in the cells that cause the root to bend? What are the signals governing the bending?

Line 33 - Symptoms- may not be a suitable word here

Author Response

Response to Reviewer 2 Comments

Point 1: Firstly, the introduction part is very short in the manuscript, maybe talk about auxin transport/gradient in root bending and development.

Response 1: Thank you for your kind suggestion. We have added sentences about auxin transport and tropic bending in the revised manuscript, “Phytohormones such as auxin regulate tropic bending [14,15], and behaviors of these hormones depending on directions of external stimuli will be a key factor to represent tropisms. In cases of auxin, PIN auxin transporters are polarly localized according to the direction of stimuli, and this asymmetric localization generates asymmetric auxin distribution [28].”

Point 2 Do authors have an idea about the gene expression of various auxin-related genes like the ones specific to auxin transporters in the coiled region? Is it an auxin-based response or other hormonal networking? What about PIN transporters and their localization during such coiling?

Response 2: Thank you for your important comments. We will obtain a part of these answers from the results of RNAseq analysis now ongoing, but not completed now. So, we cannot include these ideas in this study yet. As we answer your point 3, auxin may be related with nutritropism. We agree with your points as topics for further research. Instead, according to the definition of tropism, we have paid attention to whether the direction of nutritropic bending is differentiated from ones of other tropisms, actually gravitropism, in our original manuscript. We have added the definition of tropism in the Introduction section, “Tropism is defined as directional growth in response to a directional stimulus [14]”.

Point 3: Do authors define if it is the tip of the root or the cells adjoining the tip or above that sense the nutritropic behavior? What senses the behavior in cells? Is this behavior independent of starch sedimentation? That should be then checked by iodine staining. The bigger question here still stays then what defines or senses in the cells that cause the root to bend? What are the signals governing the bending?

Response 3: Thank you for your deep consideration. We agree that issues you pointed out are all important to reveal nutritropic mechanisms, but unfortunately, we did not do them. For example, laser ablation of columella cells may be a promising strategy to answer your first question. Similarly, we did not observe the starch sedimentation during nutritropism in this study, but our data demonstrated that the nutritropic bending direction was independent from a gravitational direction. We think that this “independence from other directional factors” is the most important issue to differentiate nutritropism from others. The third is a very tough question, and even in hydrotropism, the answer has not been obtained. We believe that ammonium receptors, which have never been discovered in plants in our understanding, will sense. The last is probably auxin. The major phytohormones to govern tropic bending are auxin or cytokinin, and previous tropism studies have shown that auxin and cytokinin regulate differential cell elongation and division, respectively. In our previous study about lateral root nutritropism, we can see differential cell elongation. So, if the mechanisms in lateral and main roots are same, auxins are signals for both nutritropism. We have now acknowledged these points and suggested some of your pointed issues as topics for further research. We have added the sentence, “Interestingly, availability of NH4+ is strongly related with auxin signaling in roots as described above. A recent study has reported that NH4+ deficiency increases auxin signaling in roots [37]. Unknown systems involved in NH4+ sensing such as NH4+ receptors might be expressed in roots, and the signal would be transduced to auxin signaling. Nutritropism should also require sensing mechanisms for NH4+ gradient, and this signal might be transduced to auxin signaling. To investigate which tissues are responsible to sense NH4+ gradient, laser-ablation is a promising technique as is the case of gravitropism research [38].”

Point 4: Line 33 - Symptoms- may not be a suitable word here.

Response 4: Thank you for your comment. We agree with you and have modified it to “adverse effects”.

Reviewer 3 Report

The authors present a follow-up study to their previous study on a novel phenomenon they coined as "nutritropism" of roots. In the reviwed manscript they present data, that the main root of rice shows tropic responses in dependance to ammonia and phosphate. Additionally, their studies revealed differential degrees of responss in dependance to their genetic background.

In this manuscript a number of experiments and data are presented, which support the existance phenomenon of nutritropism in plants and give further insight to this process. Additionally, the influence of other factors like ions (e.g. phosphate) and genetic background will be starting points for further studies.

There some minor points fo comment from my side:

  1. In lines 69-70 WRC 42 was described as "likely improper to investigate...". Although, the response of 100% appears to be artificial, te result itself cannot be neglected. What are the authors planning to do in regard of this result?
  2. In Table 1 the "Bioassay No." is given. This term is not really explained anywhere. Although, I can imagine what it stands for, I cannot see what this information is good for in this table.
  3. I would suggest to shift this table to the supplement as a documentation of raw data. Instead, I would suggest to turn it into a bar chart, which would visualize the outcomes of this experiment much better. Maybe, WRC42 should be excluded in this regard to be repeated and confirmed.

Author Response

Response to Reviewer 3 Comments

Point 1: In lines 69-70 WRC 42 was described as "likely improperto investigate...". Although, the response of 100% appears to be artificial, the result itself cannot be neglected. What are the authors planning to do in regard of this result?

Response 1: Thank you for your important comment and sorry for confusion. Owing to your comments, we now acknowledge that our discription was wrong. “Highest” is misleading. We want to say that; it is possible that too high nutritropic response, almost 100% is improper when we investigate effects of components in nutrient sources on nutritropism. It was possible that we cannnot see any positive effects with WRC 42, because it already showed the maximum response. Similarly, such too high activity might mask some negative effects. We have modified the last sentence, “because accessions with too high nutritropic response, such as WRC 42 showing 100% response, were likely improper to reveal factors which increase nutritropic response due to almost maximum scores in our quantitative method.”.  

Point 2: In Table 1 the "Bioassay No." is given. This term is not really explained anywhere. Although, I can imagine whatit stands for, I cannot see what this information is good for in this table.

Response 2: Thank you for your suggestion.As you pointed out, we should use a general word, No. of replication. We have modified this table to a bar chart as you suggested, we have used “n = “to represent numbers of replication.

Point 3: I would suggest to shift this table to the supplement as a documentation of raw data. Instead, I would suggest to turn it into a bar chart, which would visualize the outcomes of this experiment much better. Maybe, WRC 42 should be excluded in this regard to be repeated and confirmed.

Response 3: Thank you for your kind suggestion and consideration. We agree with you and have made a bar chart similar to the other figures. We leave the data of WRC 42, because we think its high response is true.

Reviewer 4 Report

This is a well written paper that describes the "Effect of Phosphate on the Activity and Sensitivity of Nutritropism toward Ammonium in Rice Roots".

The experiment was conducted using solid statical analysis and a well defined experimental design. Germplasm was selected from the NARO genebank using several WRCs accessions. 

Overall, the only minor revision I have is about the discussion. It would be beneficial to add more info about interactions between nutrients and plant hormons (i.e., IAA, ck, GA3, etc) and how they interact with nutrients to promote the nutritropism the Authors described.

Author Response

Response to Reviewer 4 Comments

Point 1: Overall, the only minor revision I have is about the discussion. It would be beneficial to add more info about interactions between nutrients and plant hormons (i.e., IAA,ck, GA3, etc) and how they interact with nutrients to promote the nutritropism the Authors described.

Response 1: Thank you for your important suggestions. Your kindly pointed topic is very interesting. Since we want to focus on the relationship between ammonium and phytohormones, we have added sentences in the Discussion section in the revised manuscript, “Interestingly, availability of NH4+ is strongly related with auxin signalling in roots as described above. A recent study has reported that NH4+ deficiency increases auxin signalling in roots [37]. Unknown systems involved in NH4+ sensing such as NH4+ receptors might be expressed in roots, and the signal would be transduced to auxin signalling.”

Round 2

Reviewer 1 Report

Although the authors have answered to the many of the comments, a very crucial aspect requires to be better considered. The more important experiment presented in this paper regards the influence on nutritropism induced by different concentrations of NH4+ and Pi (Figure 4). In this experiment, it has been tested very high concentrations of Pi (until 300 mM). In the discussion, the authors debated on the possible influence of Pi on the modulation of the nutritropism, on the relationship between this tropism and other ones as well as on the possible mechanism(s) that could be involved. In this context, we must consider the Pi concentration in the real environment (i.e. in the soil). As it is well known, [for example you can read in the chapter 1 of Phosphorus Metabolism edit by Plaxton and Lambers: “Phosphorus (P) is a pivotal nutrient for all life on Earth. It is poorly mobile in soil and inorganic P concentrations in the soil solution are <0.6 to 11 μM.”]. If, we are interested to improve our knowledge, considering the real conditions in which the plants grow, this is not a marginal aspect. We can use different experimental conditions to understand the physiological processes, but at the end, we must compare them with the reality. We can apreciate with the results proposed in the present article, but at the same time, considering the real conditions in which the plants grow in the soil, we must consider that these conditions do not occur. In conclusion, the experimental conditions adopted by the authors are really distant from the real physiological ones.

Author Response

Response to Reviewer 1 Comments

Point 1: In this context, we must consider the Pi concentration in the real environment (i.e. in the soil). As it is well known, [for example you can read in the chapter 1 of Phosphorus Metabolism edit by Plaxton and Lambers: “Phosphorus (P) is a pivotal nutrient for all life on Earth. It is poorly mobile in soil and inorganic P concentrations in the soil solution are <0.6 to 11 μM.”]. If, we are interested to improve our knowledge, considering the real conditions in which the plants grow, this is not a marginal aspect. We can use different experimental conditions to understand the physiological processes, but at the end, we must compare them with the reality. We can apreciate with the results proposed in the present article, but at the same time, considering the real conditions in which the plants grow in the soil, we must consider that these conditions do not occur. In conclusion, the experimental conditions adopted by the authors are really distant from the real physiological ones.

Response 1: Thank you for your deep consideration and interesting suggestion again. We agree with the importance to compare the conditions between experimental and natural environmental conditions.

You kindly informed us a literature, showing the Pi concentration in soil solution, < 10 µM order at most. There are problems when we compare such information with experimental conditions. In our sense, such concentrations in soil solution depend strongly on the moisture levels; e.g. when it is sunny days, the concentrations easily increase. pH conditions also affect. In fact, many studies have been conducted in 100-1000 times higher Pi (and N also) conditions at mM order in agar or hydroponic conditions, and the plants seem healthy. Oppositely, if we cultivate plants in such low nutrient conditions at 10 µM order of macronutrients in agar or hydroponics, plants do not grow healthy. This inconsistency may ask that we have to compare them carefully.

As you know, concentrations in soil only represent mean values in a macro-scale region, e.g. 100 g soil or something, and we cannot know the ranges of nutrient concentrations in a micro-scale, around newly growing root tips. Even if there are some patchy hotspots of nutrients in soil, hotspots do not increase such mean values so much. In addition, soil scientists may avoid picking up samples from such a unique area and would remove any non-soil materials from their soil samples by sieving before analysis. So, we feel unreasonable to compare concentrations in our nutrient sources and mean concentrations in real soil.

Here, we suggest that it would be better to compare contents with the reality; whether hotspots of nutrients we used occur or not in soil.

In our experimental conditions, we use plastic tubes containing 10 µL of agar media with nutrients. The molecules present in each tube are only 2 µmol in a case of 200 mM. Only a part of the contents is released from the tube, and root tips are exposed to them. At least, roots did not stop their growth, and the concentrations in root cells may be not toxic levels.

One previous report says that 1 kg fresh animal manure contains 3.6 g NH4-N and 0.7 g P, respectively. These values mean that 257 mmol NH4 and 22.6 mmol P are present in 1 kg manure. Therefore, 100 mg of animal fresh manure might be sufficient to reproduce nutrient hotspots we used in our bioassay systems.

Therefore, although no one knows the concentrations in soil solution around such nutrient hotspots, we think that similar hotspots to our experiments would occur in real soil at least. We have added the possibility of nutrient hotspots in the revised manuscript, “Nutrient sources in this study contain several µmol of nutrient molecules. According to a previous study, 1 kg of animal fresh manure contains 3.6 g NH4-N and 0.7 g P, respectively [33]. Therefore, it is possible that nutrient hotspots we applied in our bioassay systems are reproduced in natural soil environments only with 100 mg of fresh manure.”.

Reviewer 2 Report

Thank you for the response to my queries.

Author Response

Thank you for your intelligent feedback.

Round 3

Reviewer 1 Report

The actual concentration of mineral nutrients in the soil can certainly be very different, considering the different factors on which it depends. I had simply asked for a comment on the differences that inevitably exist between the experimental system adopted in this work and the real conditions that occur in a soil. The authors' response remains extremely interlocutory. Overall, this work is not very original, because it contains only some extremely preliminary evidence. I believe that a more critical analysis of the data could have made this work more interesting, even for future research.

Author Response

Thank you for all your kind suggestions. We will keep them in mind going forward.